# A Clinical Incivility Management Module for Nursing Students: A Quasi-Experimental Study

**DOI:** 10.3390/healthcare11192680

**Published:** 2023-10-03

**Authors:** Younglee Kim, Sook Young Kim, Eunhee Hong, Cheryl Brandt

**Affiliations:** 1Department of Nursing, California State University San Bernardino, 5500 University Pkwy, San Bernardino, CA 92407, USA; cheryl.brandt@csusb.edu; 2College of Nursing, Seoul Women’s College of Nursing, 38 Ganhodae-ro, Seodaemun-gu, Seoul 03617, Republic of Korea; tina@snjc.ac.kr (S.Y.K.); ehhong@snjc.ac.kr (E.H.)

**Keywords:** clinical practicum, incivility, nursing student, stress, self-efficacy

## Abstract

Background: Incivility experienced by pre-licensure nursing students in clinical settings continues to grow. Interventions for clinical incivility to nursing students are needed. Our study aimed to examine the effects of a piloted two-hour interactive incivility management module on nursing students’ perceived stress and general self-efficacy levels and preparedness for responding professionally to clinical incivility. Methods: A quasi-experimental post-test-only non-equivalent comparison design with control and experimental groups was used. Senior nursing students enrolled in a Bachelor of Science in Nursing program from a nursing college located in Seoul, South Korea, were recruited. The control group (*n* = 94) completed a self-administered online survey without the clinical incivility management module. The experimental group (*n* = 93) completed the same survey after receiving the clinical incivility management module. The two groups’ survey data were compared; qualitative data from the experimental group’s post-module debriefing session were also analyzed. Results: The prevalence of reported clinical incivility was 72.73% (*n* = 137 out of 187 participants). Clinical incivility experienced by the experimental group was significantly lower than that of the control group (z = −4.865, *p* < 0.001). However, there was no significant difference in stress levels and self-efficacy between the two groups. The mean score of the experimental group on preparedness for responding professionally to clinical incivility was statistically higher than the control group’s mean score (z = −2.850, *p* = 0.004). Conclusions: Interventions to prepare students for the experience of clinical incivility are useful; they can positively affect the students’ ability to respond professionally.

## 1. Introduction

Incivility is deviant and negative antisocial behavior [1,2] that may or may not be intended to harm others with rude, irritated, and rash behaviors, words, or gestures that lack respect for others [3,4,5]. Incivility often results in different interpretations of a situation because the intent to cause harm, as perceived by the target or observer, is unclear [6]. Recipients of incivility can experience everything from simple, mild low-intensity irritation to severe high-intensity psychological distress and perceptions of abuse, conflict, and aggression [2,3]. Psychological assault due to incivility is accompanied by harmful physical effects such as sleep disorders and digestive problems [7]. Furthermore, the experience of incivility can damage self-image [8] and yield a decrease in self-esteem [9], self-confidence [10], and life satisfaction [11]. Incivility is increasing not only in the workplace but also in human society in general [12].

Incivility is also found in teaching–learning environments; it is confirmed at an alarming rate, especially in higher education [13]. Likewise, incivility is seen in nursing education environments including classrooms and clinical areas. In particular, clinical incivility refers to the incivility that occurs in clinical settings [14]. It continues to increase among nursing students in clinical practice settings in several countries [15,16,17,18]. Clinical practice is essential to students’ preparation for becoming a nurse. However, incivility has a detrimental influence on students’ clinical practice experience, a matter of great concern to nurse educators. While applying their knowledge and skills to practice in clinical areas, nursing students experience tension, stress, psychological tiredness, helplessness [18], lack of respect and role models, and hostile and mean behaviors [15]. Nursing students also experience neglect, indifference, and unpleasant nonverbal behaviors from registered nurses (RN) [16]. Furthermore, experiencing incivility directly affects nursing care and patient safety [19]. Such speech or behavior can spread widely from individuals to an organizational system [20] with the potential to harm entire medical and healthcare organizations [21].

Stress is an outcome of experiencing incivility [22]. It can be defined as the adverse reaction people have to excessive pressure or other types of demands placed on them [23]. Some people possess characteristics that help cope with stress and adapt positively to daily life [24]. However, individuals without those characteristics may have negative reactions to stressful events [25]. Stress may inhibit their critical thinking process, interfering with their clinical performance [26].

Perceived self-efficacy is related to an individual’s expectation that they can perform the behavior(s) needed to bring about a desired outcome [27]. The strength of one’s self-efficacy influences the degree of effort an individual will expend to use their own cognitive resources, motives, or various behaviors for confronting problems [28]. Bandura [27,29] notes that self-efficacy is affected by four factors: experience with success in challenging situations, diverse life experiences, verbal persuasion, and physiological and emotional states. Of these four factors, situational experiences with successful efforts in performing difficult tasks is the most critical influence [27]. To maximize self-efficacy, it is vital to minimize outside help and increase individual efforts for success [30] within a supportive environment. Self-efficacy is an indicator of nursing students’ clinical performance [29].

Educational interventions have been developed and tested to address nursing students’ experiences of clinical incivility and to mitigate their negative effects. Tailored interventions have the potential to reduce incivility and strengthen coping methods [31]. A review of the published literature over the past 10 years (2012–2022) yielded a total of 13 studies describing interventions for incivility in nursing. Of those 13 studies, 9 focused on quality improvement projects or educational programs related to workplace incivility for nurses; only 4 studies included incivility interventions for nursing students. Of the four studies, Jenkins et al. [32] developed a monthly journal club to raise awareness of pre-licensure nursing students’ civility in a nursing program in the United States. Clark et al. [33] conducted an interactive class discussion of workplace incivility among the students of the Bachelor of Science in Nursing (BSN) program in the United States. Palumbo [34] investigated the changes in self-efficacy among students in an Associate Degree of Nursing (ADN) program in the United States using an e-module program of academic incivility. Lastly, Clark and Gorton [35] used a Cognitive Rehearsal, HeartMath, and Simulation program to improve the quality of patient care and reduce workplace incivility for pre-licensure BSN program nursing students in the United States. There is still a scarcity of studies investigating interventions addressing clinical incivility for nursing students, in particular for pre-licensure BSN nursing students.

The overall aim of our study was to add to the body of research on interventions to help nursing students anticipate, address, and cope with clinical incivility by developing and piloting a brief educational intervention.

## 2. Methods

### 2.1. Theoretical Conceptual Framework

Two theoretical frameworks provided the foundation for our study: Lazarus and Folkman’s cognitive theory of stress [36] and Bandura’s theory of self-efficacy [27]. Both frameworks have been applied to previous studies of pre-licensure nursing students’ experiences during their clinical practicum rotations. Therefore, our study measured the following key concepts/variables associated with these theories: (a) their perceived stress, in this case associated with the experience of clinical incivility, and (b) their self-efficacy.

In Lazarus and Folkman’s theory of stress [36], stress can result in either positive or negative results through a person’s coping responses to stressors. Responses to stress depend not only on individual dispositions, but also on a perceived level of threat in the environment [36]. Stressful environments can be negatively correlated with the individual learning process [37]. 

According to Bandura’s theory of self-efficacy [27], self-efficacy is an individual’s belief in personal capacity to execute the behaviors necessary to produce specific performance attainments [27,30]. Self-efficacy is a personal belief in one’s ability to control one’s actions, emotions, and motivations to solve problems, achieve goals, and complete tasks [27,38]. Higher self-efficacy is associated with greater self-motivation, commitment to learning, and overall educational success [39].

### 2.2. Specific Aims

The specific aims of this study were to: (1) investigate the relationship of clinical practicum incivility experienced by pre-licensure BSN nursing students and stress and self-efficacy levels and (2) examine the effects of an interactive incivility module on pre-licensure students’ perceived stress levels, general self-efficacy, and self-reported preparedness to respond appropriately to clinical incivility.

### 2.3. Hypotheses

I.Pre-licensure senior BSN students with more personal experiences of uncivil behavior during their clinical practicum will report higher levels of stress and lower general efficacy.II.Following participation in an interactive clinical incivility module, pre-licensure senior BSN students who complete the interactive clinical incivility module, compared to students who did not complete the clinical incivility module, will report:
a.higher self-reported preparedness to respond professionally and appropriately to incivility;b.lower perceived stress; andc.higher general self-efficacy levels.

### 2.4. Study Design

We used a quasi-experimental cross-sectional post-test-only non-equivalent comparison group research design with a self-administered online survey. The study sample was drawn from senior students enrolled in the same course in a single nursing program. Random group assignment had limited feasibility in the setting of an existing academic course; thus, previously established student cohorts within the course were assigned to either the control or the experimental group. A post-test-only design was used because of the short-term period of the study. The experimental group completed the survey on the same day as they finished the single-session clinical incivility management module; the control group completed the survey at approximately the same period but without attending the module.

### 2.5. Ethical Considerations

Institutional Review Board approval (No. 202302-HR-002) was obtained from a nursing college in South Korea. All study participants voluntarily and anonymously participated in the online survey and attended the face-to-face clinical incivility management module after being informed of the study.

### 2.6. Study Setting and Participants

Senior nursing students enrolled in a BSN program during the 2023–2024 academic year were recruited. The inclusion criteria for study participants were: (a) age of 18 years or older, (b) able to read and understand Korean, and (c) enrolled as senior nursing students in a BSN program. Potential participants who self-reported that they had been diagnosed with a mental illness by a healthcare provider or were taking prescribed medication for a mental health condition were excluded. In our study, 194 nursing students completed the survey but 7 were excluded due to their self-report of mental illness. Therefore, the data from 187 students were used for analysis. Of the 187 participants, 94 were non-randomly assigned to the control group and 93 to the experimental group (See Figure 1). The students in both groups were all taking the same nursing courses from the same instructors and were placed at the same clinical practice sites. The required sample size for our study was at least 88 in each group based on a power analysis for an alpha of 0.05, a power of 0.80, and a medium effect size of 0.50 (G*Power 3.1.9.7).

### 2.7. Instruments 

Quantitative data were collected through a self-administered 10 min online survey using five questionnaires: a Korean version of the Uncivil Behavior in Clinical Nursing Education (K-UBCNE) to measure clinical incivility experience (10 items); a Korean version of the Perceived Stress Scale (K-PSS; 10 items) for stress levels; a Korean version of the General Self-Efficacy Scale (K-GSE; 10 items) for perceived self-efficacy; a research-team-developed question about perceived preparedness to respond professionally to incivility (1 item); and a sample characteristics questionnaire (10 items). 

Students’ comments made during a debriefing session following the clinical incivility management module were collected, deidentified, and analyzed as an additional data source.

The Uncivil Behavior in Clinical Nursing Education (UBCNE) was developed by Anthony and Yastik in 2011 [40]. The initial instrument included 20 items to measure nursing students’ experiences with incivility in the clinical learning environment. Subsequent modification by Anthony et al. [41] reduced the instrument to 12 items with three subscales: Hostile/Mean (H-M; 3 items), Exclusionary Behaviors (EXBEV; 5 items), and Dismissive (DIS; 4 items). Based on the five-point Likert-type response scale (0 = never to 4 = very often) for each item, the possible scores range from 0 to 48 with higher scores representing more incivility experiences. 

The Korean UBCNE version, modified by Jo and Oh [42] for Korean populations, was used for our study. The K-UBCNE includes a total of 13 items distributed across three subscales: Exclusion (5 items), Contempt (5 items), and Refusal (3 items). The possible score range on the K-UBCNE is 0 to 52. Higher scores indicate more experiences of incivility. In the study by Anthony et al. [41], Cronbach’s alpha coefficient scores of the UBCNE ranged from 0.84 to 0.86. However, in a study of Korean nursing students, Kim et al. [43] found that Cronbach’s alpha coefficient scores of the K-UBCNE ranged from 0.78 to 0.88. In our study, the Cronbach’s alpha coefficient scores of the total K-UBCNE were from 0.77 to 0.87.

The Perceived Stress Scale (PSS) was originally a 14-item self-administered questionnaire designed to quantitatively assess how stressed individuals perceived various non-specific life situations experienced over the past month to be [44]. Subsequently, a short version of the 10-item PSS (PSS-10) was published by Cohen and Williamson [45]. In our study, the Korean version of the PSS-10 translated by Park and Seo [46] was used. Each of the 10 questions has a 5-point Likert-type response scale (0 = never to 4 = very often). Possible total scores range from 0 to 40, with higher scores indicating higher perceived stress. The Cronbach’s alpha coefficient of the PSS-10 was 0.82 from the recent study of a Pakistani population by Ashraf et al. [47]. In the Korean sample of Lee et al.’s study [48], the Cronbach’s alpha coefficient of the K-PSS-10 was 0.74; in our study, it was 0.75. 

The General Self-Efficacy Scale (GSE) was developed by Schwarzer and Jerusalem in 1995 [49]. This instrument includes 10 questions, each with a 4-point Likert-type response scale (1 = not all at true to 4 = exactly true). Possible scores range from 10 to 40; a higher score indicates more self-efficacy. In our study, the Korean version of the GSE (K-GSE) was used; this version was verified for internal consistency by Schwarzer et al. [50]. The Cronbach’s alpha coefficient of the K-GSE was 0.88 in the study by Lee et al. [51]. The Cronbach’s alpha coefficient of the K-GSE in our study was 0.87. 

A question about perceived preparedness to respond to incivility was developed by our research team after a literature review: “I feel prepared to respond professionally and appropriately when faced with uncivil behaviors in a clinical setting”. It had a 5-point Likert-type response scale (1 = Strongly Disagree to 5 = Strongly Agree). The question was piloted with five Korean nursing students to verify its clarity and readability.

Lastly, the sample characteristics questionnaire included 10 items related to recent history of mental illness, recent history of medication use for mental illness, age, marital status, education, employment status, working hours per week, religion, grade point average (GPA), and clinical incivility experiences.

### 2.8. Procedure

The intervention in our study was a research-team-developed brief clinical incivility management module that was taught during one session of the required Leadership and Management in Nursing course for senior students in the BSN program. The research team searched for both quantitative and qualitative research articles on incivility management interventions published in English and Korean. The search yielded descriptions in both quantitative and qualitative studies of incivility management modules or workshops for English-speaking nurses and nursing students [32,35,52]. However, no similar programs or workshops were found for Korean nursing students and nurses.

The incivility management module developed for Korean senior nursing students was a two-hour module, offered in a single session, with elements modeled after those described in the literature. Four senior BSN nursing student cohorts (Cohort A, B, C, and D) were selected at the chosen college for our study. Each cohort included approximately 50 students enrolled during the academic year of 2023–2024. Participants were non-randomly assigned to one of two groups: Cohorts A and B to the control group and Cohorts C and D to the experimental group. After informed consent was obtained, the control group completed the online survey without having taken the designed incivility management module; the experimental group took the online survey after completing the module. Half the students in the experimental group completed the module in a morning session; the other half completed it in an afternoon session on the same day. Participation in the survey and the incivility management module was voluntary.

A principal investigator (PI) and a co-principal investigator (Co-PI) attended the experimental group’s sessions of the Leadership and Management in Nursing course as quests. The PI, a guest lecturer, taught the incivility sessions to each cohort of the experimental group, while the Co-PI attended the sessions to assist. The PI implemented the module as follows:(1)Video Presentation: Background to Incivility—Students watched a 20 min video explaining topics including general company or organizational system management and describing examples of uncivil behaviors (20 min).(2)Small Group Discussion #1—The students were subdivided into small groups of five or six students to discuss what they saw and felt about the video and then wrote a short discussion summary (20 min).(3)Live Researcher Presentation: Incivility Impact and Management—The PI described research on incivility definitions, incivility’s prevalence in nursing, incivility’s effects on nursing education and practice, professional responses to incivility, and incivility management and prevention (30 min).(4)Small Group Discussion #2—The students held a 20 min group discussion to share individual experiences of incivility from their clinical sites and develop a professional response to clinical incivility based on the knowledge learned from the module (20 min).(5)Group Discussion Summary—Representatives of each small group briefly presented members’ descriptions of their experiences with clinical incivility and their proposed professional response (20 min).(6)Debriefing—The students and the research team held a debriefing session to share what the students learned from the clinical inactivity management module (10 min). Members of the research team wrote field notes on students’ comments.(7)Online Survey—The students were asked to complete the online survey after the informed consent forms.

Students in the control group completed the online survey on the same day; students in the experimental group completed the module and online survey a week later on the same day of the week. All data collection and intervention for our study were performed between March and April 2023. After completing the online surveys, members of both the control group and the experimental group received an electronic gift card in a token amount.

### 2.9. Quantitative Analysis

For statistical data analysis, SPSS version 28 was utilized. Descriptive statistical tests (e.g., frequency counts and distribution) were used to analyze interval-level data about sample characteristics and incivility, stress, and self-efficacy scores as well as self-reported preparedness to respond professionally to incivility. According to the results of the Shapiro–Wilk test of the K-UBCNE, the K-PSS, and the K-GSE, the sample was not normally distributed. Therefore, nonparametric statistics (i.e., Chi-Square, Mann–Whitney U test, and Spearman correlation) were used. The sample characteristics of the control and experimental groups were compared to determine group equivalence. 

To address Hypothesis I, the control and experimental groups’ mean scores on the K-UBCNE (clinical incivility), the K-PSS (perceived stress), and the K-GSE (general self-efficacy) were calculated and compared. A Spearman correlation analysis of the strength of relationships among the total scores on the K-UBCNE, the K-PSS, and the K-GSE was performed. To address Hypothesis II.a, II.b, and II.c, a Mann–Whitney U test was conducted to determine whether there was a difference between the two group scores on the single survey item related to perceived readiness to respond to clinical incivility, the K-PSS, and the K-GSE.

### 2.10. Qualitative Analysis of Debriefing Session Field Notes

Recognizing the potential value of experimental group members’ comments during the debriefing session, the research team used a thematic analysis [53] to analyze the field notes taken during the debriefing session. The process of theme analysis included reading students’ comments, identifying meanings and themes, reviewing the themes found, and defining the themes.

## 3. Results

### 3.1. Sample Characteristics

Survey data from 187 senior students enrolled in a BSN program from a nursing college located in Seoul, South Korea, were analyzed. There were no statistically significant differences in the sample characteristics between the control (*n* = 94) and experimental group (*n* = 93) (see Table 1). The mean ages of the control and experimental groups were 23.73 (SD = 3.21) and 23.47 (SD = 2.68).

### 3.2. Experience of Clinical Incivility

Of the 187 participants, 136 (72.73%) reported that they experienced incivility from classmates (*n* = 39, 20.9%), patients (*n* = 46, 24.6%), patients’ families (*n* = 30, 16%), nurses (*n* = 97, 51.9%), physicians (*n* = 17, 9.1%), and staff or others (*n* = 21, 11.2%) at their clinical sites. 

Table 2 showed the Mann–Whitney U test results for the K-UBCNE, the K-PSS, and the K-GSE between the control and the experimental groups. Interestingly, the Mann–Whitney U test indicated that the K-UBCNE score was significantly lower in the experimental group (*Md* = 34, *n* = 93) compared to the control group (*Md* = 43, *n* = 94), *U* = 2571.5, z = −4.865, *p* < 0.001, with a large effect size *r* = 0.36. The results of the Mann–Whitney U test for the subscales of the K-UBCNE also showed that Exclusion (*U* = 2663.5, z = −4.624, *p* < 0.001), Contempt (*U* = 2799.0, z = −4.258, *p* < 0.001), and Refusal (*U* = 2799.0, z = −4.258, *p* < 0.001) each scored significantly lower than the control group.

### 3.3. Stress and Self-Efficacy 

Table 1 showed that there was no statistically significant difference between the K-PSS scores of the two groups (z = −1.561, *p* = 0.118); Hypothesis II.b was not supported.

The median rank of the K-GSE for the control group (*Md* = 27, *n* = 94) was lower than the experimental group (*Md* = 29, *n* = 93). However, there was no statistically significant difference between the two groups; Hypothesis II.c was not supported. 

### 3.4. Self-Reported Preparedness to Respond Professionally to Incivility 

In the results, the Mann–Whitney U test for the two groups on the item about professional response to incivility showed that the experimental group (*Md* = 104.5, *n* = 93) was significantly higher than the control group (*Md* = 83.45, *n* = 94) with U = 3548.0, z = −2.850, *p* = 0.004. Thus, Hypothesis II.a was supported.

### 3.5. Correlation Computation

Bivariate Spearman correlations were calculated to investigate the relationships between the total score on the K-UBCNE, the total score on the K-PSS, and the total score on the K-GSE (See Table 3). The total score on the K-UBCNE was positively correlated with the total score on the K-PSS, (*r_s_*  = 0.369, *n* = 187, *p  *< 0.001). Thus, Hypothesis I was partially supported. The greater the experience of clinical incivility, the higher the reported stress.

### 3.6. Findings of Debrief Session

Through the thematic analysis of the researchers’ field notes following the above-described process, five themes were identified: (a)Needing more time to learn about and discuss clinical incivility. A student’s comment exemplifying this theme was “It was too short to learn and discuss clinical incivility today. I hope I have more time next time”.(b)Learning how to professionally handle incivility for their clinical practicums. Students used phrases such as “I learned” and “I knew”.(c)Recognizing the importance of clinical incivility for nursing education and the profession. One student captured the thoughts of the group by saying, “I thought clinical incivility was a simple problem, but I found it so important for nursing education and profession”.(d)Taking time to express personal negative feelings about incivility from clinical sites. One student said, “Today was a time to express negative feelings about incivility I experienced in the clinical sites”.(e)Forming empathy with other students over the experience of clinical incivility. A student’s comment exemplifying this theme was “I was surprised that other students were experiencing clinical incivility just like me”.

## 4. Discussion

In our study, the prevalence of clinical incivility among Korean nursing students was 72.73% (*n* = 136). Hong et al. [20] and Kim et al. [43] reported prevalence rates of clinical incivility among nursing students in South Korea of 51.9% (*n* = 97) and 91.46% (*n* = 375), respectively. A recent study by McDonald et al. [54] of a sample of Canadian nursing students found that 70% (*n* = 182) experienced clinical incivility, especially in acute care settings. Considering our findings and those of other investigators, it cannot be emphasized enough that the need to address clinical incivility in nursing students is great. 

According to the statistical analysis of the K-UBCNE in our study, the scores on the Exclusion, Contempt, and Refusal subscales were all significantly higher in the control group than in the experimental group. In addition, the experimental group participants reported significantly fewer experiences of clinical incivility than the students in the control group did. We have not encountered this finding in our review of the literature. One possible explanation for this finding is that the experimental group completed the module that included a definition of incivility and research on it. This may have affected their understanding and subsequent reporting of their clinical incivility experience. The total score of the K-UBCNE showed a significant positive correlation with students’ perceived stress levels. This finding is similar to other relevant studies [43,55]. This finding, of course, speaks to correlation but not causation. 

Interestingly, many students in our study (*n* = 97, 51.9%) experienced clinical incivility from nurses. This issue has been reported in other studies [55,56,57]. In particular, nursing students in South Korea mentioned that at the clinical areas, nurses (a) neglected nursing students during their clinical practicum, (b) treated them like nursing assistants, and (c) did not respect them; this negatively affected students’ motivation to enter the nursing profession [58]. Nurses from clinical sites who work with students have an important role in providing a high-quality clinical education [59], helping students prepare for clinical practice. Nurses should effectively support nursing students [56,60] and serve as models of professional nursing ethics [61]. Nurses must adjust their attitudes to see nursing students not as annoying but as important to the future of the healthcare system [62]. Furthermore, nursing schools and healthcare organizations should intentionally assist nurses to understand the extent of the problem of incivility among nursing students and its negative consequences.

The students in the experimental group of our study did not, as hypothesized, report lower perceived stress nor higher general self-efficacy compared to the control group after completing the clinical incivility module. Importantly, the experimental group’s self-rated preparedness to professionally respond to clinical incivility was significantly higher than that of the control group. The relatively brief clinical incivility module had a positive effect on the students’ perceptions of being ready to manage clinical incivility when they encountered it. Clark and Gorton [35] investigated the effects of a 150 min incivility intervention based on Cognitive Rehearsal, HeartMath, and Simulation on 146 pre-licensure nursing students in the United States. A total of 120 students (82%) could address incivility situations and 129 (88.4%) could reduce their stress levels after the intervention. Jenkins et al. [32] conducted a mixed method study of a journal club intervention on a sample of U.S. pre-licensure nursing students’ experience of incivility (*n* = 25). Jenkins et al. [32] found that the students could address nursing incivility, refused to participate in uncivil behaviors, and were able to prevent or manage incivility after the intervention. In a pre-test/post-test study using an e-learning module about incivility with 110 nursing students from an associate degree nursing program degree in the United States, Palumbo [34] found students’ levels of self-efficacy increased significantly pre- and post-test (*p* < 0.05). Our clinical incivility module may need refinement to have the desired impact on students’ stress and self-efficacy levels, but our results indicate it had a positive impact on their preparedness to manage incivility. 

Finally, during the debriefing session, students in the experimental group reported that they gained a greater understanding of the impact of clinical incivility in nursing education and the profession as a whole. They reported having learned how to respond professionally to clinical incivility after completing the module, but noted they needed additional time to learn about and address the issue. 

Our study findings support the impact of even a brief clinical incivility module on the students’ perceptions of preparedness to respond professionally when encountering clinical incivility. However, more relevant research is needed.

Recently, many quantitative and qualitative studies on the experiences of clinical incivility among nursing students in South Korea have been conducted [15,43,56,58,63,64]. However, little research exists regarding clinical incivility management interventions for pre-licensure nursing students in South Korea. Our study protocol and findings can guide nursing educators and researchers who wish to develop and evaluate interventions in clinical incivility, especially for pre-licensure BSN nursing students in South Korea. Furthermore, our findings prompt us to make the following suggestions related to clinical incivility experienced by pre-licensure nursing students.

(a)Nursing ethics and nursing leadership courses in BSN programs should include a module on the management of clinical incivility.(b)Further research on the most appropriate timing, length, format, and content for incivility management programs is recommended. Our module was brief (two hours), involved senior nursing students, and included the above-described content and interactive components. Earlier intervention, alternate formats, and/or additional content may have a greater impact on students’ stress and self-efficacy levels and other important outcomes.

The fact that our study was conducted with senior students at one nursing college is a limitation; caution must be used in generalizing to all nursing students in South Korea and in other countries. In light of the characteristics of the course that served as the setting for the intervention, as well as the short time period for the study, we used a cross-sectional post-test-only non-equivalent (non-randomized) comparison group design. To increase internal validity, future studies should use a longitudinal pre-test/post-test design with student participants randomized into control and experimental groups. To evaluate the impact of the clinical incivility management module, the measurement of key variables should occur at baseline prior to the delivery of the module and again after students in both groups have completed additional clinical practice. Measures of stress and self-efficacy more specific to the experience of clinical incivility would likely have greater ability to discriminate between general life stress and self-efficacy and the stress of and self-efficacy for addressing clinical incivility. Notably, our study used primarily quantitative methods; the lived experiences of students encountering clinical incivility was not investigated. Qualitative research strategies are needed to investigate the personal experiences of clinical incivility and the effects on students.

## 5. Conclusions

Clinical incivility is often found among nursing students, including from practicing nurses in clinical areas. It is important for nurses and nurse educators to recognize the prevalence and negative impacts of clinical incivility experiences among nursing students. It is also urgent to develop interventions for incivility management for nursing students. Our study reaffirmed the problem of clinical incivility in nursing students and piloted a brief one-time incivility management intervention. Our findings can inform a larger effort to raise awareness of clinical incivility and ameliorate its damaging effects, especially for pre-licensure BSN nursing students.

## Figures and Tables

**Figure 1 healthcare-11-02680-f001:**
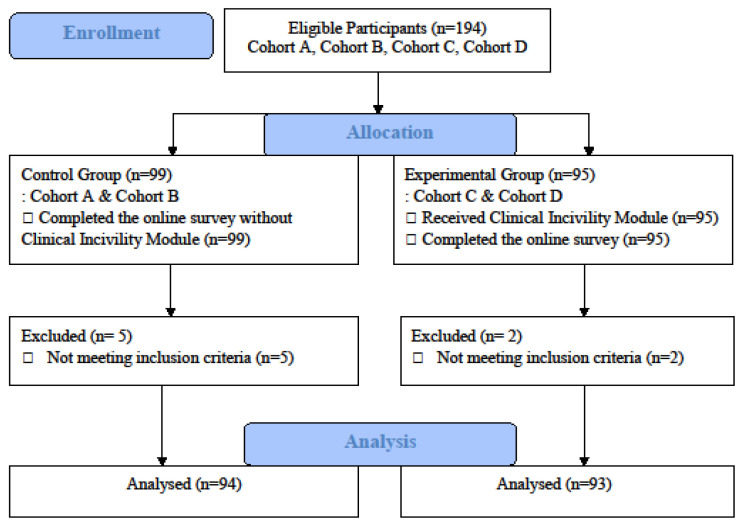
A flowchart of participant recruitment and retention.

**Table 1 healthcare-11-02680-t001:** Results of chi-square test for sample characteristics (*n* = 187).

Variable	Control Group	Experimental Group	*x* ^2^
	*n* (%)	*n* (%)	
	94 (50.27)	93 (49.73)	
Age			0.876
<25	69 (73.40)	70 (75.27)	
≥25	25 (26.6)	23 (24.73)	
Marital status			0.621
Single/Without partner	93 (98.94)	92 (98.92)	
Married/Living with partner	1 (1.06)	1 (1.08)	
Education			0.132
Unemployed	90 (95.74)	91 (97.85)	
Employed	4 (4.26)	2 (2.15)	
Religion			0.102
No	62 (65.96)	50 (53.76)	
Yes	32 (34.04)	43 (46.24)	
Catholic	12 (12.50)	13 (14.00)	
Protestant	18 (18.50)	25 (26.90)	
Buddhist	2 (3.04)	4 (4.30)	
Other	0 (0.00)	1 (1.04)	
GPA			0.444
<3.0	2 (2.13)	4 (4.30)	
≥3.0	92 (97.87)	89 (95.70)	

Note: GPA; grade point average.

**Table 2 healthcare-11-02680-t002:** Mann–Whitney U test of nursing students’ clinical incivility, perceived stress, and general self-efficacy (*n* = 187).

Instrument	Group (*n*)	Mean Rank	z	Sig
K-UBCNE	Control (94)	43.0	−4.865	<0.001
	Experimental (93)	34.0		
Exclusion	Control (94)	18.0	−4.624	<0.001
	Experimental (93)	13.0		
Contempt	Control (94)	17.0	−4.258	<0.001
	Experimental (93)	13.0		
Refusal	Control (94)	8.0	−4.120	<0.001
	Experimental (93)	6.0		
K-PSS	Control (94)	28.0	−1.561	0.118
	Experimental (93)	28.0		
K-GSE	Control (94)	27.0	−1.062	0.228
	Experimental (93)	29.0		

Note: K-UBCNE; Korean version of Uncivil Behavior in Clinical Nursing Education, K-PSS; Korean version of the Perceived Stress Scale, K-GSE; Korean version of the General Self-Efficacy Scale.

**Table 3 healthcare-11-02680-t003:** Intercorrelations among the study variables (*n* = 187).

	K-UBCNE	K-PSS	K-GSE
K-UBCNE	1		
K-PSS	<0.369 *	1	
K-GSE	0.073	−0.138	1

Note: K-UBCNE; Korean version of Uncivil Behavior in Clinical Nursing Education, K-PSS; Korean version of the Perceived Stress Scale, K-GSE; Korean version of the General Self-Efficacy Scale. * *p* < 0.001 (two-tailed).

## Data Availability

The data presented in this study are available from the corresponding author upon reasonable request.

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
