# Peer review of "A Clinical Incivility Management Module for Nursing Students: A Quasi-Experimental Study"

_healthcare, 2023, doi:10.3390/healthcare11192680_

Round 1

Reviewer 1 Report

Thank you for the opportunity to review this paper. The aim of the study A Clinical Incivility Management Module for Nursing Students: A Quasi Experimental Study. Clinical incivility refers to disrespectful, rude or unprofessional behavior that occurs in healthcare settings. It includes actions such as demeaning comments, condescension, verbal abuse and other forms of negative interaction. Situations involving rudeness among healthcare professionals negatively affect patient care. Poor communication and str ained relationships between nursing staff

members and students can lead to errors, reduced teamwork and decreased patient satisfaction. Continuous education and training in interpersonal and conflict resolution skills should be provided to both nursing stu dents and practicing nurses to help prevent rudeness and manage conflict constructively. The need to level unfavorable behavior during student practice, and mitigate its harmful effects, is a very important factor affecting later interpersonal behavior and relationships. 

However, there are major issues that need to be resolved before the study is accepted for publication.

1. Procedure module students of the selected university and older students were included in the study please clarify;

2. What stre ssful situations did the students indicate?

3. No discussion in the results of the statistics of rude behavior (which ones) line 221;

4. No discussion of the results from the student session of the five key topics from the student discussion line 315

5. The sample size is small for a study that aimed to assess whether there is a relationship between personal experiences of clinical rudeness and their impact on students.

Author Response

Manuscript Sections

Reviewer 1

Thank you very much for taking the time to review this manuscript. Please find the detailed responses below and the corresponding revisions/corrections highlighted (Blue) in the re-submitted file.

5. The sample size is small for a study that aimed to assess whether there is a relationship between personal experiences of clinical rudeness (KUBCNE) and their impact on students (K-PSS). 

Author Response:

We would love to had a larger sample, of course, but the required sample size for our study was at least 88 in each group based on a power analysis for an alpha of .05, power of .80 and a medium effect size of .50 (G*Power 3.1.9.7). 

Abstract

Introduction

Method - Theoretical

Objective/Aims/Hypotheses

Design

Setting & Participants

Instruments

Procedure

Analytical/statistical methods

Qualitative analysis

Results

1. Procedure module students of the selected university and older students were included in the study please clarify;

Author response:

Clarification of the study participants was added in the Results section.

2. What stressful situations did the students indicate? 

Author response:

The K-PSS instrument did not have items asking about specific stressful situations. Instead, it asked about general situations and whether they were deemed stressful. Due to the study design, a quantitative design, our survey had no questions to ask specific stressful situation. It can be one of the study limitations.

3. No discussion in the results of the statistics of rude behavior (which ones) line 221

Author Response:

We interpret this comment to be suggesting that we should add some detail after Table 2 about the item(s) [uncivil behaviors] from each of the K-UBCNE subscales that the Control group experienced more frequently than the Experimental Group. The data of the subscales of K-UBCNE were added.

Discussion

4. No discussion of the results from the student session of the five key topics from the student discussion line 315 

Author response:

The themes found in the students' debriefings were briefly described, and exemplar comments included, in the Discussion.

Reviewer 2 Report

Thank you for your submission- incivility faced by nursing students is a significant issue internationally.  The manuscript is well written.  I have two questions that I believe need to be addressed in the manuscript:

1. What is the relationship of the authors with the participants? I believe positionality of the authors is needed.

 2. Why was random assignment not used?  How did you choose to place students into groups if random assignment was not used? 

Author Response

Manuscript Sections

Reviewer 2

Thank you very much for taking the time to review this manuscript. Please find the detailed responses below and the corresponding revisions/corrections highlighted (Blue) in the re-submitted file.

Abstract

Introduction

Method - Theoretical

Objective/Aims/Hypotheses

Design

Setting & Participants

Instruments

Procedure

 2. Why was random assignment not used?  How did you choose to place students into groups if random assignment was not used? 

Author Response:

It was not easy to use the randomization in a limited time and system in the BSN program.  Therefore, it is one of the limitations in our study.

1. What is the relationship of the authors with the participants? I believe positionality of the authors is needed. 

Author response:

A PI and a Co PI had no information of the study participants.

Analytical/statistical methods

Qualitative analysis

Results

Discussion

Reviewer 3 Report

Dear authors

I consider that the study is relevant, novative and useful because interventional research allows the evolution of knowledge in nursing.

To my opinion your investigation can be enhanced, correcting some aspects in your paper:

- Introduction: define the concept of incivility

- Study design and discussion: explain the reason why you chose not to carry out a pre-test evaluation moment. This aspect also needs to be developed in the discussion because it compromisses your results.

- quantitative analysis: before using t tests you need to verify the sample normality because the study is not randomized. O tales 1 and 2 you must apply the Shapiro-Wilk test and if the sample is not normal, use non parametric tests.

This investigation can be reconsider after major revision (method and results should be clarified).

Author Response

Manuscript Sections

Reviewer 3

Thank you very much for taking the time to review this manuscript. Please find the detailed responses below and the corresponding revisions/corrections highlighted (Blue) in the re-submitted file.

- Method and results should be clarified (general summary).

Author response:

We addressed your specific comments: the Introduction, study design, statistical analysis, and discussion (see below).

Abstract

Introduction

- Introduction: define the concept of incivility

Author response:

The definition of incivility was added

Method - Theoretical

Objective/Aims/Hypotheses

Design

- Study design and discussion: explain the reason why you chose not to carry out a pre-test evaluation moment. This aspect also needs to be developed in the discussion because it compromises your results.

Author response:

Additional detail was added in both the Study Design and the Discussion (limitations) sections.

Setting & Participants

Instruments

Procedure

Analytical/statistical methods

- Quantitative analysis: before using t tests you need to verify the sample normality because the study is not randomized. O tales 1 and 2 you must apply the Shapiro-Wilk test and if the sample is not normal, use non parametric tests.

Author response:

Non-parametric statistics were utilized to analysis the survey data

Qualitative analysis

Results

Discussion